# From Adhesion to Invasion: Integrins, Focal Adhesion Signaling, and Actin Binding Proteins in Cervical Cancer Progression—A Scoping Review

**DOI:** 10.3390/cells14201640

**Published:** 2025-10-21

**Authors:** Marta Hałas-Wiśniewska, Patryk Zawadka, Wioletta Arendt, Magdalena Izdebska

**Affiliations:** Department of Histology and Embryology, Faculty of Medicine, Collegium Medicum in Bydgoszcz, Nicolaus Copernicus University in Toruń, Karłowicza 24, 85-092 Bydgoszcz, Poland; mhalas@cm.umk.pl (M.H.-W.); patrykzawadka@cm.umk.pl (P.Z.); warendt@cm.umk.pl (W.A.)

**Keywords:** cervical cancer, actin binding proteins, focal adhesion kinase, focal adhesion proteins, integrin, progression, metastasis, adhesion

## Abstract

Background: Cervical cancer (CC) is one of the most common malignancies in women worldwide. Its progression involves a cascade of processes, including proliferation, migration, invasion, and metastasis. Each stage is regulated by specific signaling pathways. Objective: This scoping review aimed to map current evidence on the role of cell adhesion-related molecules, including integrins, focal adhesion (FA) proteins, and actin-binding proteins (ABPs), in CC progression. These protein groups act in a coordinated manner—integrins perceive and transmit extracellular matrix (ECM) signals, FA proteins mediate intracellular signaling, and ABPs reorganize the cytoskeleton, ensuring the continuity of adhesion and motility processes. Methods: A structured literature search was conducted for studies published between 2015 and 2025. Eligible articles described the role of adhesion-related proteins in migration, invasion, or EMT in CC. Data were synthesized thematically according to protein families. Results: The evidence highlights integrins, FA/FAK, and ABPs as interconnected regulators coordinating ECM signaling and cytoskeletal remodeling during CC progression. Their dysregulation is associated with enhanced migration, EMT induction, angiogenesis, and therapy resistance. Conclusions: This review provides a unique, integrated perspective linking adhesion molecules with invasion mechanisms in CC progression, providing new insights into their interplay. Understanding the interaction between these proteins is therefore a crucial step in the treatment of CC and may facilitate the discovery of biomarkers and support the development of targeted therapies.

## 1. Introduction

Cancer progression is a complex biological process involving numerous changes at the molecular, cellular, and tissue levels. The characteristic uncontrolled proliferation of cells, as well as their ability to migrate, invade, and metastasize, significantly affects the course of the disease and patient survival. Proteins involved in specific signaling pathways play a crucial role in these processes. They enable cancer cells to respond to various signals from the extracellular matrix (ECM) and adapt their motile and invasive properties [1,2].

One of the key steps in cancer progression is the loss of normal cell adhesion and the remodeling of cellular interactions with the ECM and surrounding microenvironment [3,4]. The transition between adhesion and invasion is particularly critical in cervical cancer (CC) [5]. Changes in adhesive elements, such as integrins and focal adhesion proteins, not only determine cell adhesion to the ECM but also initiate signaling cascades that reprogram the cytoskeleton and the cell’s metabolic and transcriptional response [6,7,8]. Through the remodeling of adhesion sites and the formation of specialized structures, such as invadopodia and podosomes, cancer cells acquire the ability to proteolytically degrade the ECM and penetrate adjacent tissues [9]. This directly links adhesion with the invasive and metastatic phenotype [10]. In CC, where the main initiating factor is infection with the highly oncogenic human papillomavirus (HPV), viral oncoproteins can further influence the expression and function of integrins and adhesion proteins, accelerating the transition from adhesion dysfunction to aggressive invasion [11,12]. These conditions favor tumor cell detachment and cancer progression, while perturbations in the expression of focal adhesion components lead to cytoskeletal reorganization and activation of signaling pathways promoting epithelial–mesenchymal transition (EMT), angiogenesis, and treatment resistance [13,14].

Integrins are key components of this signaling network. They constitute a group of transmembrane adhesion proteins whose primary role is to sense signals from the external environment. It is important to note that integrins also participate in intracellular signaling, which influences their conformation, ligand binding affinity, and adhesive activity. This bidirectional communication—both external and internal—is essential for the dynamic regulation of cell adhesion and the cell’s response to changing microenvironmental conditions. Their crucial role, therefore, encompasses not only cell adhesion to the substrate but also initiating signaling cascades responsible for proliferation, migration, and invasion [15]. Another key step in cancer progression is the processing of information into an intracellular response. This function is attributed to focal adhesion protein (FAs) and focal adhesion kinase (FAK) [16,17]. The literature details the influence of these molecules on the activity of signaling pathways such as extracellular signal-regulated kinase (ERK) and akt signaling pathway (AKT), as well as on biological processes such as proliferation, migration, and EMT [18,19]. Effectors in cancer progression include actin binding proteins (ABPs), which directly interact with the cellular cytoskeleton. These proteins enable cells to change shape, acquire motor capabilities, and thus penetrate surrounding tissues [20].

The multidimensional interaction of integrins, FAs, FAK, and ABPs creates a continuous signaling chain—from signal reception from the ECM, through its transduction, to a tangible outcome such as cell migration or invasion. Dysfunction of any element of this complex puzzle can lead to exacerbation of neoplastic features and, consequently, accelerated cancer progression. This general mechanism is conserved across multiple types of cancer, including CC.

CC is a challenging disease worldwide, being one of the most common malignancies in women, second only to breast, colon, and lung cancers. It is associated with significant morbidity and mortality. Almost all cases are caused by human papillomavirus (HPV) infection. Extensive research over the years has improved our understanding of HPV, resulting in its classification into high-risk and low-risk types. High-risk HPV infection accounts for the majority of cases [21,22]. However, genetic, environmental, and immunological factors also contribute to disease development. The time between the precancerous stage to invasive carcinoma is typically slow. This creates opportunities for early detection and comprehensive preventive measures. Unfortunately, despite advances in diagnostics, vaccine development, and numerous treatment options (surgery, radiotherapy, chemotherapy), CC remains a serious problem in low- and middle-income countries, and the prognosis for patients with metastatic disease remains poor. Therefore, research into understanding the molecular mechanisms involved in the development and progression of CC remains a pressing issue [23]. Identifying new prognostic markers and therapeutic targets offers hope for improved patient outcomes [21].

Given the expanding body of research on adhesion-related signaling in CC, a comprehensive synthesis of evidence is needed to clarify how these molecular systems contribute to tumor progression. This scoping review aims to map and summarize current knowledge on the roles of integrins, focal adhesion and focal adhesion kinase proteins, and actin-binding proteins in the progression, migration, and invasion ofCC. This review considers proteins that constitute a specific signaling axis of CC (Table 1, Figure 1). By integrating findings from *in vitro*, *in vivo*, and bioinformatic studies, this review highlights their potential as prognostic markers and therapeutic targets.

## 2. Methods

### 2.1. Protocol and Reporting

Although the protocol for this review was not formally registered, it was developed a priori based on the PRISMA-ScR model. The research question, inclusion and exclusion criteria, and search strategy were defined before the literature review began to ensure methodological transparency and reproducibility. This review was conducted based on the checklist and guidelinesof the PRISMA Extension for Scoping Reviews (PRISMA-ScR) recommended by Tricco et al. [24].

Given the heterogeneity of the molecular mechanisms and experimental models examined across studies, a scoping review approach was chosen to comprehensively map and summarize the current evidence.

### 2.2. Eligibility Criteria

Results presented in this review were chosen based on a meticulous methodology and detailed insight into the influence of integrins, FAs, FAK and ABPs on the CC progression. Some research had to be excluded in order to maintain the accessibility and applicability of this study. The main exclusion factors were date of publication older than ten years as of the moment of writing this review (2015–2025), inaccessibility to the study, duplication, and lack of relevance to the topic. Only peer-reviewed original articles and reviews were included. Conference abstracts, editorials, and non-scientific reports were excluded. No language restrictions were applied; however, the search yielded only studies published in English. Included articles granted the most up-to-date information and conclusions, shedding light on the processes of migration and invasion of CC. The abundance of data allows researchers to continue studying chosen proteins in order to understand the complex malignancy that CC is.

### 2.3. Search Strategy and Information Sources

In the selected studies, we focused on the relationships between proteins and their impact on the potential of cancer cells to migrate and invade other sites. We focused primarily on recent publications available in the PubMed database, published between 2015 and 2025. The last literature search was conducted in August 2025. We searched for relevant publications using precise search terms that included “cervical cancer” and the name of the selected protein. No language restrictions were applied, but only full-text articles in English were included. In the section presenting individual proteins, we focused on the most recent publications, while in the introductory section, some cited works were more outdated. The oldest publication we cited was published in 2003 (Burd et al.), and the most recent in 2025. Incorporating primarily recent studies ensures the validity of the presented data and allows us to identify which issues are already understood and which require further research. We first screened publications by title and abstract, looking for links between integrins, FAs, FAK, and ABPs and CC migration and invasion. This allowed us to narrow the selection to publications that were irrelevant. We then narrowed the publication timeframe to the last ten years to access the most up-to-date information. Finally, the articles were carefully read and reviewed for relevant details.

### 2.4. Selection of Sources of Evidence

Two reviewers independently screened titles and abstracts for relevance. Full-text screening was conducted for potentially eligible studies, and disagreements were resolved through discussion and consensus.

### 2.5. Data Charting Process

A structured data extraction table was developed to capture the following information:Protein/Target;Key role/Function in CC;Main mechanisms/Pathways;Clinical/Experimental insights;References.

Tables with the collected data are included at the end of each section.

### 2.6. Data Items

For each included study, the following items were collected: (1) bibliographic information (authors, year of publication); (2) study design and experimental model (*in vitro*, *in vivo*, clinical, or bioinformatic analysis); (3) the specific adhesion-related protein(s) investigated (integrins, focal adhesions, focal adhesion kinases, actin-binding proteins); (4) associated signaling pathways and molecular mechanisms; (5) key findings related to cell adhesion, migration, invasion, and epithelial–mesenchymal transition; and (6) reported clinical or translational relevance (e.g., prognostic value or therapeutic potential). Data were extracted independently by the reviewers to ensure accuracy and consistency.

### 2.7. Synthesis of Results

Extracted data were synthesized using a descriptive and thematic approach. The studies were grouped according to the main protein categories involved in CCprogression—integrins, FAs, FAK, and ABPs. Within each category, findings were organized based on their reported impact on key biological processes, including cell adhesion, migration, invasion, and EMT.

Results were summarized in narrative form and supported by summary tables highlighting study design, molecular mechanisms, and clinical implications. This scoping review aimed to map the existing evidence rather than to estimate effect sizes.

### 2.8. Visualization of Protein Expression in CC

Although not part of the scoping review methodology per se, complementary analyses were performed using publicly available transcriptomic datasets (TCGA, GSE63514, GSE67522) to visualize the expression of key adhesion-related proteins (integrins, FA/FAK, and ABPs) in CC versus normal tissues. The data were used to visualize general expression. Differences in expression levels between groups (cancer vs. normal tissue) were assessed using parametric tests (unpaired *t* test with Welch’s correction) and nonparametric tests (Mann–Whitney U test) after assessing the sampling distribution (Shapiro–Wilk test). The significance level was set at 0.05, and the analysis was performed in GraphPadPrism 8.

## 3. Results

The study selection process is summarized in the PRISMA-ScR flow diagram (Figure 2).

### 3.1. Integrins in CC

Integrins are a group of transmembrane heterodimers composed of α and β subunits. They are essential for maintaining proper interactions and the flow of biochemical/mechanical signals between cells and the external environment, including the ECM and cytoskeleton [6,15]. To date, 19 α and 8 β subunits have been identified, which can form over 20 distinct αβ heterodimers with different ligand-binding and functional specificities. Importantly, the biological activity of integrins depends on the specific pairing of α and β subunits, which determines their affinity for ECM components. Integrins influence, adhesive properties and play an important role in the progression of various types of cancer [7,17]. In the case of CC, integrins α3β1, α5β1, and αvβ6 have been shown to contribute to cancer progression and they are often associated with processes such as migration, invasion and EMT. This review focuses on three integrin subunits (α3, α5, and β1), which, according to current research, demonstrate particular importance in CC progression. They represent key components of the integrin heterodimers most frequently disrupted in this cancer. By providing an overview of the current knowledge, we aim to better understand how integrin-mediated signaling contributes to migration, invasion, angiogenesis, and treatment resistance in CC [15,25,26,27].

A total of 29 studies investigating the role of integrins in CC progression were included in this review. These studies utilized a variety of experimental approaches, including *in vitro* cell line models, *in vivo* assays, immunohistochemistry, and bioinformatic analyses of publicly available datasets. Overall, the literature highlights that specific integrin subunits—particularly ITGA3, ITGA5, and ITGB1—are frequently implicated in the regulation of cell adhesion, migration, invasion, EMT, and angiogenesis inCC.

The following sections provide a detailed summary of the available evidence for each integrin subunit, describing their molecular functions, signaling pathways, and clinical or experimental insights. Corresponding data are presented in Table 2, which organizes the key findings in a concise and systematic manner. Expression patterns of integrins across CC and normal tissues are further illustrated in Section 3.1.4, integrating results from both published studies and transcriptomic datasets.

#### 3.1.1. Integrin α3

The laminin-binding integrin (LBI) group, including integrin α3 (ITGA3), is crucial for cell adhesion. Harryman’s team studied various cancer types to analyze copy number alterations (CNAs) or mutations in the five-gene LBI signature. They reported that CNAs, including integrins α3 and β4, across multiple cancer type, also CC, were significantly associated with poor overall survival and therapy response [28]. In turn, Du et al. determined the role and mechanism of ITGA3 in the progression and angiogenesis of CC. Immunohistochemistry (IHC) analyses revealed high levels of ITGA3 in tissue samples, which correlate with poor prognosis among patients. The obtained results showed that ITGA3 promotes the migration and invasion of CC cells, secretion Matrix Metalloproteinase 9 (MMP-9), and consequently induces angiogenesis, with the most important mechanism of action being the Src/ERK/FAK pathway [29]. Tan et al. confirmed elevated ITGA3 expression in CC. Results demonstrated that manipulation of ITGA3 levels induced an increase proliferation, migration, and EMT via PI3K/AKT signaling based on both experimental and bioinformatic analyses (TCGA, CESC, GEO) [25]. Furthermore, siRNA knockdown of ITGA3 and ITGA2 levels in CC cell lines to confirm the role of integrins in inhibiting the migration and invasion and was correlated with decreased mucin 1 (MUC1) levels and suggesting regulation by ERK phosphorylation [30].

#### 3.1.2. Integrin α5

Integrin α5 (ITGA5) is of particular interest in the case of CC. Several studies have reported an association of high ITGA5 levels with a poor prognosis in patients with CC [27,31,32,33]. Liu et al., based on bioinformatic analysis of methylation-related genes from TCGA and GEO datasets revealed that low expression level and hypermethylation of ITGA5 was associated with better survival in cervical squamous cell carcinoma (CSCC). The research team hypothesized that both methylation status and expression level of ITGA5 correlate with prognostic significance in patients with CC [31].

The literature describes the importance of ITGA5 in angiogenesis, the formation of new vessels that facilitate tumor development, also in CC [15,27,34,35]. Xu et al. showed that ITGA5 knockdown limits the angiogenic capacity of CC cells, partly by regulation of vascular endothelial growth factor (VEGF) and fibronectin (FN1). They also noted a positive correlation between microvessels density and ITGA5 levels [27]. Similar observations were made by Li et al. who identified ITGA5 among 15 angiogenesis-related key genes associated with immune infiltration. Additionally, they found link between ITGA5 and immune cells, particularly activated mast cells, tumor-associated macrophages (TAMs), and Th2 and M2 cells [36]. In line with the above results, Wu et al. confirmed ITGA5 overexpression in CC tissues. Furthermore, they incorporated it into five-gene prognostic model predicting patient survival [37].

Additional *in vitro* studies on CC cell line models have shown that ITGA5 promotes the proliferation, migration, and invasion. Yao et al. indicated that reduced integrin levels significantly inhibit CC cell migration and invasion and may also be the subject of a prognostic model with accurate predictions for patient therapy [38]. Further evidence was presented in the study by Zhou et al. The team linked ITGA5 activity to metastatic potential and tumor growth through IMP3-HK2 axis [32].

Molecular analyses conducted in the context of various CC treatments revealed an association between ITGA5 levels and the treatment response [39,40,41,42]. Chuang et al. described the use of microRNA-128 as a potential treatment and hypotheses that it inhibits the motor and adhesion properties of CC cells by reducing transcript expression of molecules such as ITGA5 [41]. Another method was nano-niosomes with *Anabasis setifera* extract which was described by Ebadi et al. The researchers observed, reduction in wound healing capacity in HeLa cells and also reduction in ITGA5 levels by *Anabasis setifera* extract treatment [40].

#### 3.1.3. Integrin β1

Another subunit of integrin found to be overexpressed in CC is β1 (ITGB1)—a protein involved in cell adhesion, interactions with the ECM, and regulation of cell migration [1,36,43,44]. The available literature indicates correlations between ITGB1 expression and CC tumor stage, grade, and HPV-related risk. Yi et al. identified key genes and pathways responsible for the progression of CC, and observed that ITGB1 levels were overexpressed, particularly in the most advanced disease stage (IV phase), confirming its role in promoting metastasis. Their study also revealed an interaction between integrin, FN1 ligand, and MMP9, which could contribute to ECM degradation and enhanced CC migration. Moreover, co-expression and survival analyses confirmed that elevated ITGB1 levels correlate with poorer overall survival (OS) in patients [1].

Another study their analyzed RNA from peripheral blood of CC patients to identify genes distinguishing high-grade squamous intraepithelial lesions (HSILs) from low-grade squamous intraepithelial lesions (LSILs). Among the 10 genes examined, ITGB1 expression was found to be lower in HSILs compared to LSILs. Zou et al. suggested that reduced ITGB1 levels together with downregulation of Rho GTPase activating protein 18 (ARHGAP18) may play a role in the loss of cell adhesion and promote progression from a milder to a more aggressive stage of CC [45].

In turn, Meng et al., reported that ITGB1 is upregulated in HPV-positive CSCC and associated with poor OS [46]. Similar observations were described in a study by Kwon et al. who concluded that high expression levels of molecules such as ITGB1 and ITGA5, correlated with reduced OS and may contribute to therapy resistance [26].

ITGB1 has also been analyzed in the context of its molecular target and the pathways associated with it. Zhang et al. showed that the miR-183-5p suppresses metastasis by targeting ITGB1 [43]. Similar observations were made by Yang et al., who revealed that miR-361-5p targets hsa_circ_CSPP1 and ITGB1 likely through the PI3K-Akt pathway. This axis significantly inhibited tumor growth *in vivo* and cell proliferation and migration while increasing the rate of apoptosis [47]. Li et al., reported a positive correlation between hypoxia-inducible factor-1A (HIF-1A) and ITGB1, and their influence on CC progression, which makes the HIF-1A/ITGB1 axis a potential therapeutic target in CC. Moreover, high HIF-1A expression was correlated with poorer prognosis [48]. In turn, Liu et al. examined the role of retinoic acid-binding protein 2 (CRABP2) in CC and found that its silencing induced apoptosis and inhibited cell migration, while inhibition of ITGB1 or FAK/ERK pathway reversed CRABP2-driven promigratory effects. These findings suggest that CRABP2 may act through the HuR-ITGB1/FAK/ERK signaling axis [49].
cells-14-01640-t002_Table 2Table 2Summary of studies investigating integrin family members in cervical cancer.Protein/TargetKey Role/Function in CCMain Mechanisms/PathwaysClinical/Experimental InsightsReferencesITGA3Promotes migration, invasion, angiogenesisSrc/ERK/FAK, PI3K/AKT,ERK phosphorylationHigh expression correlates with poor prognosis; targeted by siRNA; affects EMT and MUC1 regulation[24,27,28,29]ITGA5Drives proliferation, migration, invasion, angiogenesisVEGF/FN1,IMP3-HK2,TGF-βHigh expression correlates with poor prognosis; modulated by microRNA-128 & Anabasis *setifera*; included in prognostic models[15,26,31,32,33,34,35,36,37,38,39,40,41]ITGB1Regulates adhesion, metastasis, response to hypoxiaFN1/MMP9,PI3K-Akt,HIF-1A/ITGB1, FAK/ERKOverexpressed in advanced CC and HPV-positive cells; targetable via miRNAs or CRABP2 inhibition; correlates with worse OS and therapy resistance[1,25,35,42,43,44,45,46,47,48]

#### 3.1.4. Expression Patterns of Integrins in CC

Integrins play a significant role in the development of CC. They are believed to regulate cell interactions with the ECM and activate pathways promoting migration, invasion, and angiogenesis. ITGA3 supports these processes primarily by stimulating the Src/ERK/FAK and PI3K/AKT pathways. ITGA5 is particularly strongly associated with angiogenesis and its impact on the tumor microenvironment. High levels of this protein are associated with poorer prognosis and shorter patient survival. ITGB1 has a more complex nature—in many studies, its overexpression accompanies progression and metastasis, although early stages of the disease yield ambiguous results. A common element of these integrins’ action remains the activation of the FAK/ERK and PI3K/AKT axes, which may represent an attractive therapeutic target. There are no reports regarding ITGAV in CC—another integrin that plays a significant role in signaling the cell/ECM axis.

Multiple studies have reported integrins upregulation in CC tissues and cell lines compared to normal cervical epithelium. Given the involvement of integrins in cancer progression, they also become potential therapeutic targets in CC.

To extend the evidence summarized in this review, publicly available transcriptomic datasets (TCGA, GSE63514, GSE67522) were examined to map the expression trends of selected integrin genes (ITGA3, ITGA5, ITGAV, ITGB1) across normal and CC tissues (Figure 3).

The integred data visualization illustrates a consistent pattern of elevated expression. These differences were particularly evident for ITGA3 and ITGB1, where the highest levels of statistical significance were observed in independent datasets (Figure 3). These findings align with the reviewed literature, supporting the hypothesis that dysregulated integrin signaling contributes to CC progression and may represent a promising avenue for targeted therapeutic strategies.

### 3.2. Adhesive Proteins and Focal Adhesion in CC

Across the literature, 26 studies investigating FAs and FAK in CC were included in this scoping review. These investigations, using cell lines, animal models, and computational analyses, demonstrate that FA/FAK components are central regulators of cell adhesion dynamics, cytoskeletal remodeling, and signaling pathways that drive migration, invasion, and EMT. The sections below provide a comprehensive overview of the key FA proteins and FAK, summarizing their molecular functions, associated signaling pathways, and experimental or clinical insights. Table 3 and Table 4 organize the findings across studies, while Sections Expression Patterns of Focal Adhesion in CC and Expression Pattern of FAK in CC integrate data on expression patterns and pathway interactions to visualize their role in CC progression.

#### 3.2.1. Focal Adhesion Proteins

FAs are a group of proteins that constitute a specific bridge between integrins and the cytoskeleton, influencing the dynamics of changes in cell behavior. Proteins regulate the formation and disintegration of focal junctions, thus controlling the polarity, migration, and invasiveness of cancer cells [50]. In the context of CC, FAs constitute a particularly important link between the loss of normal cellular adhesion and the acquisition of invasive capacity. Changes in the expression or activity of key FA components—such as FAK, paxillin (PXN), or talin (TLN)—are associated with increased cell motility, EMT, and treatment resistance, highlighting their significant role in CC progression. Studies published in recent years indicate that modulation of FA activity may be an attractive target for anticancer therapies, especially when combined with FAK inhibitors or targeted interventions on selected adhesion proteins [16,17].

##### Talin

TLN, a cytoskeletal protein, links integrins to actin filaments. It influences cell adhesion and migration. Dysregulation of TLN1 expression is associated with cancer progression, promoting invasion, metastasis, and treatment resistance [51].

Ma et al. investigated the functional role and possible regulatory mechanisms of circRNA_400029 in CC cell lines and found that it affects TLN1 expression via the circRNA_400029/miR-1285-3p/TLN1 axis. Expression analyses showed that circRNA_400029 and TLN1 level were significantly elevated in CC tissue and cancer cells, while miR-1285-3p was downregulated. Functional assays confirmed that miR-1285-3p directly targets TLN1. Moreover, downregulation of TLN1 level reduced CC cell proliferation. *In vivo* studies further confirmed that silencing circRNA_400029 or TLN1 suppressed tumor growth. These findings highlight the importance of TLN1 and circRNA_400029/miR-1285-3p/TLN1 pathway in CC progression [52].

##### Vinculin

VCL is a significant component of FAs and ABPs. Literature indicates that increased protein expression promotes the stabilization of adhesive properties and limits cell migration potential in various cancer types [53]. The team led by You demonstrated that exosomal miR-663b targets the 3′-UTR of VCL mRNA, leading to decreased protein expression and promoting angiogenesis in Human Umbilical Vein Endothelial Cell (HUVEC) cells. Analyses of CC tissue and TCGA data confirmed reduced VCL expression in comparison to normal samples. Moreover, *in vivo*, intratumoral delivery of miR-633b suppressed VCL expression and inhibited tumor growth by limiting angiogenesis [54]. Du et al. examined the response of the HeLa cell line to three commonly used chemotherapeutic agents: cisplatin, doxorubicin, and 5-fluorouracil. Team revealed that above mentioned cytostatic agents decreased cell adhesion in dose- and time-dependent manner, corresponding with reduced level of VCL. Interestingly, low concentrations of cytostatics showed an increase in VCL intensity, which the authors explained as a mechanism of cell resistance through cell stabilization and changes in adhesion [55]. Another study reported that combination of 5-fluorouracil and magnol synergistically inhibited proliferation, adhesion, and invasion of CC cells. The observed changes in protein levels, including VCL, lead to cytoskeletal remodeling and, consequently, changes in shape and migratory properties [56].

##### Paxillin

PXN is a focal adhesion protein widely implicated in the growth and metastasis of various types of cancer. Its most important functions include participation in proliferation, motility, metastasis, and tissue and ECM reorganization through interactions with integrins. In CC, elevated PXN levels have been reported compared to normal tissues, which correlated with invasive potential and greater colony-forming capacity of cells. Gu et al. suggested that the protein could be identified as a prognostic factor among CC patients [57].

Interestingly, the relationship between PXN and anchorage-independent cell growth in CC cells is complex. Yoo et al. demonstrated that tyrosine-phosphorylated Cool-related protein 1 (Cat-1) interacts with PXN to modulate anchorange-independent CC cell growth via Akt signaling. The authors suggested that PXN may promote ADP-Ribosylation Factor 1 (Arf1) activation and thus indirectly influence the functioning of the mTOR/Akt pathway [58].

Das and Maiti further showed that laminar shear stress induces autophagy-dependent PXN turnover in HeLa cells, enhance migration and invasion while increasing secretion of pro-invasive factors such as interleukin 6 (IL-6) and active matrix metalloproteinase 2 (MMP-2) [59].

Recent studies have also highlighted natural compounds as potential therapeutic agents targeting the FAK/PXN pathway [16,60,61,62,63]. An example is apigenin, a natural flavonoid found in fruits, vegetables, and herbs, which exhibits potent anticancer activity. The results obtained by Chen et al. described how apigenin inhibits cell viability, induces G2/M cycle arrest and mitochondrial apoptosis, and limits migration and invasion by silencing the FAK (including PXN and ITGB1) and PI3K/AKT signaling pathways [61]. Similarly, combined treatment with luteolin and asiatic acid synergistically blocked proliferation and migration of CC cells via inhibition of the FAK/PXN, PI3K/AKT, and JNK/p38 MAPK pathways and activation of ERK signaling [62]. Arctigenin, a lignan derived from *Arctium lappa* L. reduced proliferation and invasion of HeLa and SiHa cells through inhibition of the FAK/PXN pathway, while FAK overexpression reversed this effect [63]. Genistein, another natural flavonoid, also suppressed HeLa cell viability, adhesion, and migration, downregulating FAK, PXN, and EMT-related genes such as Snail and Twist [60,64]. The above literature reports indicate that PXN and the associated FAK/PXN pathway may be an important molecular target in CC therapy.
cells-14-01640-t003_Table 3Table 3Summary of studies examining focal adhesion (FA) proteins in cervical cancer.Protein/TargetKey Role/Function in CCMain Mechanisms/PathwaysClinical/Experimental InsightsReferencesTalin (TLN)Supports adhesion, motility, metastasisIntegrin activation, focal adhesion signalingUpregulated in CC; silencing decreases migration and invasion[52]Vinculin (VCL)Stabilizes focal adhesions, promotes motilityIntegrin-actin linkage, focal adhesion stabilizationOverexpressed in CC; knockdown reduces migration and invasion[53,54,55]Paxillin (PXN)Promotes proliferation, migration, invasionFocal adhesion regulation, actin cytoskeleton remodeling, FAK-PXN-MAPK signalingOverexpressed in CC; knockdown reduces aggressive phenotypes[16,56,57,58,59,60,61,62,63]

##### Expression Patterns of Focal Adhesion in CC

FAs play an important role in CC cell migration and invasion. The literature cited above indicates that their functions are not uniform. For example, TLN1 and PXN typically promote an aggressive cell phenotype by increasing invasive properties, whereas VCL more often acts to stabilize adhesion and, in some contexts, limit migration. The strongest evidence points to the importance of the FAK/PXN pathway as a potential therapeutic target, regulated by both molecular factors (miRNAs, circRNAs) and natural compounds with anticancer activity. At the same time, significant research gaps remain in the literature. In particular, data on the role of zyxin (ZYX) in CC—a linker between the ECM and actin filaments in focal adhesions—are lacking. A significant limitation is also the number of clinical studies assessing the prognostic significance of FA proteins.

To complement these findings, publicly available transcriptomic datasets were examined to map the expression patterns of selected FA genes (VCL, TLN1, PXN, and ZYX) in normal and cancer samples (Figure 4). The comparative analysis revealed that TLN1 and ZYX expression were significantly decreased in CC tissues in independent cohorts. For VCL and PXN, the observed differences were less pronounced and dataset dependent. These differences may be due to the available dataset size and the chosen platform. Nevertheless, the results suggest the clinical relevance of selected proteins, highlighting their role in altering CC cell adhesion (Figure 4).

#### 3.2.2. Focal Adhesion Kinase

Focal adhesion kinase (FAK) is an enzyme that forms a central signaling axis responsible for transducing mechanical and chemical signals from integrins into cellular responses. This non-receptor tyrosine kinase is located cytoplasmically in cells with the ability to interact with ECM or other cells, and is responsible for their adhesion, migration and invasion. Furthermore, FAK-related signaling promotes loss of epithelial characteristics and EMT [16].

There are numerous reports in the literature that FAK overexpression and activation are associated with enhanced metastatic potential in cancer cells [65]. The team led by Chen analyzed the clinical significance of FAK in CC. Results indicated a statistically significant, four-fold increase in FAK expression in tumor material compared to normal tissue. Furthermore, the use of siRNA against FAK reduced proliferation and migration and sensitized the cells to cytotoxic drugs [16].

FAK in CC may be regulated by various proteins. The SAM and SH3 domain-containing 1 (SASH1) overexpression significantly inhibited cell invasion and proliferation by downregulation of MMP-2 and MMP-9, suggesting its tumor-suppresive role via the FAK-related axis [66]. Another protein associated with activation of the FAK pathway in CC was secreted protein acidic and rich in cysteines-like 1 (SPARCL1). Zhang et al. demonstrated that protein suppressed CC cell proliferation and migration through the FAK/ERK pathway, while osteopontin overexpression increased p-FAK levels [67]. LIM domain kinse 1 (LIMK1) promoted CC progression through oxidative stress and the Src-dependent FAK pathway, activating actin and cofilin-1 (CFLN-1) [68]. Moreover, CRABP2 and Eukaryotic Translation Initiation Factor 3 Subunit D (EIF3D) were found to promote tumor growth by activating FAK through the Integrin β1/FAK/ERK and GRP78-related pathways, respectively [49,69].

Natural compounds also modulate FAK signaling in CC. Bufalin inhibited proliferation, migration, and invasion of CC cells by downregulating FAK and α2/β5 integrins, with enhanced effects when combined with paclitaxel [70]. Shikonin and parthenolide reduced CC cell viability and motility through suppression of FAK/AKT/GSK3β signaling and inhibition of EGF-induced FAK phosphorylation [71,72]. Tretinoin decreased FAK expression and metastasis both *in vitro* and *in vivo* by directly binding to key FAK residues [73]. Similar inhibition of the FAK/PXN axis was reported for other natural agents, including luteolin with asiatic acid, apigenin, and arctigenin [61,62,63,64].
cells-14-01640-t004_Table 4Table 4Summary of studies evaluating Focal Adhesion Kinase (FAK) expression and signaling in cervical cancer.Protein/TargetKey Role/Function in CCMain Mechanisms/PathwaysClinical/Experimental InsightsReferencesFAKPromotes proliferation, migration, invasion, metastasisIntegrin β1/FAK/ERK; Src-dependent FAK; actin/cofilin; FAK/AKT/GSK3βOverexpressed in CC tissues; silencing reduces migration/proliferation and sensitizes cells to drugs[64]FAK/SASH1Tumor suppressor regulating FAKFAK-related signaling axis; downregulation of MMP-2/9SASH1 downregulated in CC; overexpression inhibits invasion and proliferation[65]FAK/SPARCL1Suppresses CC proliferation/migrationFAK/ERK axisSPARCL1 overexpression reduces CC cell proliferation and migration; osteopontin increases p-FAK[66]FAK/LIMK1Promotes cytoskeletal reorganization and CC progressionSrc-dependent FAK; actin/cofilin activation; oxidative stressLIMK1 enhances invasive potential Via FAK-mediated cytoskeletal changes[67]FAK/CRABP2Promotes CC progressionIntegrin β1/FAK/ERK axisSilencing CRABP2 reduces p-FAK, p-ERK, and ITGB1; inhibits migration and invasion[68]FAK/EIF3DPromotes CC progressionFAK activation via GRP78Overexpression activates FAK; knockdown inhibits tumor growth in xenograft model[48,68]FAK/Natural compoundsTherapeutic inhibition of FAK-driven proliferation and migrationFAK/integrin/AKT/GSK3β pathwayBufalin, shikonin, parthenolide, tretinoin, luteolin, etc., reduce proliferation/migration/invasion *in vitro* and *in vivo*; combination with paclitaxel enhances effect[60,61,62,63,69,70,71,72]

##### Expression Pattern of FAK in CC

The presented above studies demonstrate the importance of the FAK enzyme and its associated signaling pathways. This kinase regulates fundamental biological processes such as proliferation, migration, and invasion, which collectively contribute to cancer progression. Its activity can be regulated by various regulatory proteins as well as numerous natural and pharmacological compounds. These findings highlight FAK as a promising therapeutic target in multiple types of cancer, including CC.

To complement these findings, expression data of the gene encoding FAK (PTK2) were extracted from publicly available transcriptomic datasets (TCGA, GSE63514, GSE67522). As shown in Figure 5, expression levels were significantly elevated in CC tissues compared to normal material. This trend was observed in all three cohorts. This observation supports the involvement of FAK signaling in the molecular landscape of (Figure 5).

### 3.3. Actin Binding Proteins in CC

Cancer cell proliferation, migration, and invasion are accompanied by rapid cytoskeletal reorganization, the growth of migratory protrusions, and the formation of stable focal contacts. This would not be possible without the involvement of numerous ABPs, which regulate actin polymerization through nucleation, elongation, severing, and capping, as well as modifying filament organization through crosslinking and bundling, e.g., stress fibers [20]. Through the coordinated action of ABPs such as CFL, PFN, FSCN, and VCL, cells can form lamellipodia, filopodia, and invadopodia, which are essential for acquiring motor skills [74]. In CC, low expression or activation of specific ABPs has been associated with enhanced cytoskeletal remodeling, EMT, and increased metastatic potential Furthermore, several ABP proteins act as downstream effectors of integrin-FAK signaling, translating biochemical signals from focal adhesions into mechanical forces that drive cell motility. Alterations in these proteins not only promote invasion but may also contribute to treatment resistance by altering cell stiffness and adhesion dynamics. Collectively, ABP proteins constitute a key component of the integrin-FAK axis, mediating cytoskeletal and mechanical responses that underlie CC progression [60,62,63].

ABPs have been extensively studied for their role in regulating cytoskeletal architecture and enabling cancer cell motility and invasion. A total of 60 studies examining ABPs in CC were analyzed. Research encompassing *in vitro*, *in vivo*, and bioinformatic approaches indicates that dysregulated ABP activity contributes to enhanced migration, EMT, and metastatic potential in CC.

The following subsections summarize the current evidence for specific ABPs, highlighting their molecular mechanisms and potential clinical relevance in CC progression. Key results are presented in Table 5 (at the end of the section) with supporting visualizations in Section 3.3.10, providing a clear overview of ABP-mediated regulation of cell adhesion and invasion in CC.

#### 3.3.1. Actinins

Alpha-actinins (ACTNs) belong to the spectrin supergroup and are ABPinvolved in cytoskeletal organization and cell mobility. They influence cell viability as well as tumor migration and invasion capacity [75,76].

Recent research has shown that levels of ACTN4 are elevated in various cancers, including CC [74]. Ma et al. noted that elevated levels of ACTN4 in the serum of CC patients correlated with FIGO stage, lymph node metastasis, and lymphovascular invasion. The team concluded that ACTN4 levels could be a useful diagnostic and prognostic marker despite limited sample size [75]. Similar conclusions were stated by the team led by Zhu, who found increased ACTN4 levels on both serum and cervical tissue of Cervical Intraepithelial Neoplasia 3+ (CIN3+) patients compared to the control group. Researchers also noted that the mRNA of ACTN4 was overexpressed in CC tissues, particularly in advanced FIGO stage, larger tumors, and lymph node metastases. Combined with SCC-Ag, ACTN4 showed strong diagnostic and prognostic potential for CIN3+ cases [77].

Jung et al. demonstrated that ACTN4-knockdown in SiHa and HeLa cells, resulted in a reduction of proliferation and stem cell sphere formation in comparison to control cells. Additional, ABCG2, a chemoresistance marker regulated via the Wnt/β-catenin pathway, was decreased after downregulation of ACTN4, indicating that ACTN4 could serve as potential therapeutic target [76].

Several proteins and anticancer drugs disrupt ACTN overexpression by blocking pathways in which it could play a role, mainly Wnt/β-catenin, EMT, and PI3/AKT/mTOR pathways [56,78,79]. Chen et al. found that a combination of 5-fluorouracil and magnolol inhibited cell proliferation via induction of cytoskeletal and morphological changes. It lowered cell adhesion, migratory and invasive abilities of cells, and the expression of the PI3K/AKT/mTOR pathway, which decreased the frequency of EMT. The combination of 5-fluorouracil and magnolol decreased the expression of ACTN, and the results were significantly better than using 5-fluorouracil alone [56]. The obtained research results indicate a significant involvement of ACTN in the invasion and migration of CCcells.

Huseinovic et al. identified ACTN1 as a target of the tumor-suppressive miR-129-5p, which, when restored, reduced cancer cell migration, invasion, and angiogenesis. Overexpression of miR-129-5p decreased ACTN1 levels and cell viability, confirming its regulatory role [78]. Li and Wang noted that Epidermal Growth Factor Receptor Antisense RNA 1 (EGFR-AS1) is overexpressed in CC and that knockdown on SiHa and CaSki cell lines caused reduced cells’ proliferative, migratory, and invasive abilities while heightening the rate of apoptosis. The researchers also noted that ACTN4 initiates the WNT pathway and silencing EGFR-AS1 suppressed ACTN4 expression and inhibited Wnt signaling, linking these molecules in cervical carcinogenesis [79]. In turn, Wang et al., found that reduced Na+/H+ exchanger regulatory factor 1 (NHERF1) expression in invasive CC correlated with elevated ACTN4 and F-actin polymerization, suggesting that NHERF1 inhibits migration and invasion by downregulating ACTN4 [80,81]. Additionaly, An et al. also demonstrated that ACTN4 promotes EMT via Akt-dependent Snail upregulation, enhancing MMP-9 expression and thereby cell migration and invasion [82].

#### 3.3.2. Cofilin

Cofilin (CFL), an ABP, regulates F-actin depolymerization and cell–matrix adhesion. Its activity is modulated by the protein phosphatase slingshot homolog (SSH1) and LIM kinases LIMK1 and LIMK2 [83].

Proteomic analysis by Pappa et al. revealed that CFL1, a key actin-depolymerizing factor, was consistently upregulated in CC lines compared to controls [84]. Given its role in F-actin turnover and its association with cell migration and proliferation, this upregulation underscores CFL1’s potential involvement in CC pathogenesis.

Jia et al. demonstrated that LIMK1, which phosphorylates and inactivates CFL, was significantly overexpressed in CC tissues and promoted tumor growth, migration, and invasion via a ROS/Src-dependent pathway involving p-FAK, p-ROCK1/2, and p-CFL1. Inhibitors of ROS or Src reversed these effects, confirming the pathway’s role in actin cytoskeleton remodeling [68].

A research team led by Du identified stratifin (SFN) as another regulator of actin dynamics. SFN was upregulated in CC tissues compared to normal cervix, and functional assays demonstrated that protein promotes proliferation, migration, and invasion of CC cells while inhibiting apoptosis. Crucially, SFN overexpression enhanced F-actin accumulation and cytoskeletal reorganization through increased phosphorylation of LIMK2 and CFL, wheras SFN knockdown reduced p-cofilin expression and disrupted actin dynamics. The authors described SFN as a positive regulator of CFL activity via the LIMK2/p-CFL axis, reinforcing the central role of ABP in CC progression [85].

Hezinger et al. reported that Nod-like receptor protein NOD1 (NOD1), a pattern-recognition receptor upregulated in HPV-driven CC, increased cell migration independently of its canonical NF-κB or RIPK2 pathways. NOD1 interacted with the actin-associated protein HCLS1-Associated Protein X-1 (HAX-1), and its expression reduced p-CFL levels, suggesting that protein mediates NOD1-driven motility while HAX-1 acts through a distinct mechanism [83]. Team led by Mercier investigated the effects of two novel anti-mitotic compounds, ESE-15-one and ESE-16, on microtubule dynamics and the actin cytoskeleton in HeLa cells. Both compounds increased CFL phosphorylation, leading to actin stabilization and reduced cell migration and invasion, indicating their potential anti-metastatic effects. The observed rise p-CFL, known to inhibit actin-severing activity, corresponded with stabilization of actin stress fibers. These findings led the researchers to implicate cofilin phosphorylation as a key mediator of actin remodeling in response to ESE compound treatment in CC cells, contributing to impaired motility and invasive behavior [86]. Wang et al. demonstrated that extracellular vesicles carrying miR-146a-5p promote CC metastasis via the WWC2/YAP axis. YAP activation suppressed cofilin phosphorylation, enhanced F-actin depolymerization, and facilitated metastasis. Conversely, YAP silencing increased p-cofilin and reduced F-actin turnover [87]. Additional studies showed that miR-509-3p enhanced apoptosis and chemosensitivity through the RAC1/PAK1/LIMK1/cofilin pathway, while miR-29a downregulated PAK1, p-LIMK, and p-cofilin, inhibiting CC cell proliferation, migration, and invasion [88,89].

#### 3.3.3. Cortactin

Cortactin, otherwise known as cortical ABP, mobilizes Arp2/3 proteins and binding to actin microfilaments. It plays key roles in forming lamellipodia and invadopodia, cell motility, endocytosis, cell death, and tumors’ invasive abilities. High cortactin expression has been observed in multiple cancers, including CC [90]. Bumrungthai et al. focused on examining cortactin, p16INK4A, Ki-67, and HPV E6/E7 ribonucleic acid (RNA), which could be promising diagnostic biomarkers. Immunostaining revealed a progressive increase in cortactin expression from normal tissue to LSIL, HSIL, and squamous cell carcinoma (SCC), with the strongest cytoplasmic staining observed in SCC cases. The authors proposed mathematical models incorporating patient age and biomarker levels to predict cervical lesion progression [90]. Cheng et al. demonstrated that cortactin expression is regulated by the VEGF-C/miR-326 signaling pathway in CC. In clinical samples, VEGF-C was upregulated while miR-326 was downregulated, showing an inverse correlation. *In vitro*, VEGF-C suppressed miR-326 expression through c-Src signaling, leading to increased cortactin levels and enhanced invasion of SiHa cells. This effect was reversed by cortactin knockdown or miR-326 overexpression, indicating that VEGF-C promotes CC cell invasion by downregulating miR-326 via c-Src, thereby relieving repression of cortactin [91].

#### 3.3.4. Diaphanous-Related Formin 3

Diaphanous-related formin 3 (DIAPH3) is ABP involved in regulating actin and microtubule networks during key cellular processes like meiosis, mitosis, and intracellular transport. It modulates GSK3β and β1-integrin signaling to stabilize microtubules and control actin assembly, thereby influencing cell adhesion and migration [92]. While DIAPH3 has been studied in different types of malignancies, its exact role and mechanisms of action in CC are still poorly understood [93,94].

Wan et al. identified DIAPH3 as a potential oncogene in CC, with overexpression correlating with poor prognosis across TCGA, GEPIA, and HPA datasets. They reported that *in vitro*, DIAPH3 knockdown in HeLa and SiHa cells suppressed proliferation, colony formation, and tumor growth in mice, accompanied by mTOR pathway inactivation. Moreover, authors noted reduced p-AKT, mTOR, and p-p70s6k, alongside increased PTEN expression [93]. Similarly, Chen et al. confirmed a high-level expression of DIAPH3 in CC tissue and linked it to worse outcomes and survival for patients. Gene set enrichment analyses further associated DIAPH3 overexpression with activation of MYC, Tumor growth factor β (TGF-β), and angiogenesis pathways, and inverse correlation with p53 and inflammatory signaling [94]. Both groups also explored the immune context of DIAPH3 expression. Elevated DIAPH3 was positively correlated with Th2 and NK cells, but negatively with B cells, macrophages, dendritic cells, and Tregs [93,94]. Moreover, DIAPH3 expression inversely linked with immune checkpoint molecules including CTLA4, PDCD1, HAVCR2, LAG3, and TIGIT, suggesting potential link between DIAPH3 and immune evasion mechanisms in CC [94].

#### 3.3.5. Ezrin/Radixin/Moesin

The Ezrin-Radixin-Moesin (ERM) family consists of these three closely related proteins, which act as a scaffolding to crosslink their target transmembrane proteins to actin filaments and facilitate their localization to the plasma membrane [95]. Kobori et al. reported Ezrin (EZR) as crucial for CD47 surface localization in CC cells, showing direct EZR–CD47 interaction and co-upregulation in tumor tissues, whereas Radixin (RDX) and Moesin (MSN) lacked this association [95].

Multiple studies have highlighted EZR’s overexpression in CC and its role in promoting tumor aggressiveness [96,97,98]. Carvalho et al. found EZR upregulated in CC, correlated with negatively with disease-free and disease-specific survival [96]. Fadiel et al. observed EZR levels increasing along the progression from normal epithelium through CIN to invasive carcinoma, while estrogen receptor expression declined [97]. Li et al. confirmed EZR overexpression in CIN and CC, with higher levels linked to poor differentiation, advanced stage, deep stromal invasion, lymph node metastasis, and reduced overall survival [98]. High EZR often co-occurs with low E-cadherin and high p16^INK4A/Ki-67, marking disease progression [99,100]. Adittionaly, EZR knockdown suppresses proliferation, migration, invasion, and EMT in HeLa and SiHa cells through PI3K/Akt downregulation, while enhancing epithelial morphology [101,102,103]. Interestly, Xu et al. reported that quercetin potentiates cisplatin cytotoxicity partly by inhibiting EZR [104].

EZR regulation involves multiple upstream effectors: EBP50 overexpression decreases CC cell motility via ERM interaction [105,106]. Mubthasima et al. reported that EPHA2 knockdown decreased EZR and CD133 expression, impairing CC cells motility and colony formation [107]. Lectin Galactoside-Binding Soluble 1 (LGALS1) enhances EZR-mediated invasiveness, whereas sevoflurane and estradiol increase EZR expression and pro-migratory activity [86,107,108].

Although EZR is the most studied ERM member, RDX and MSN modulate to CC biology. A c-Jun/HIF1A-AS2/miR-34b-5p/Radixin axis promotes proliferation and invasion [109], while MSN facilitates PD-L1 membrane localization, supporting immune evasion [110]. EZR similarly anchors PD-L1 at the plasma membrane, and its phosphorylation correlates with PD-L1 stability [111]. Clinically, high EZR expression, perinuclear localization, and promoter methylation serve as potential biomarkers of poor prognosis. RDX and MSN, though less studied, emerge as key modulators of immune checkpoint localization and downstream signaling, warranting deeper investigation.

#### 3.3.6. Fascin

Fascin (FSCN1), as ABP, involved in filopodia formation and cell motility, is strongly associated with cervical carcinogenesis. Ghalejoogh et al. detected FSCN in all CC samples, with high expression in 65% of tumors and strongest association in HPV16-positive cases, suggesting HPV-driven upregulation [112]. These findings suggest a potential link between HPV infection, particularly type 16, and FSCN upregulation during cervical tumorigenesis. Additionally, Tian et al. confirmed that HPV16 E6/E7 oncogenes stabilize FSCN1 mRNA via the METTL14/IGF2BP3 axis, enhancing malignancy and correlating with poor prognosis [113].

Similar results were obtained by Csizmár et al., who showed FSCN expression increases from LSIL to SCC and correlates with p16^INK4A and Ki-67, linking it to cell proliferation and disease progression [114]. These findings suggest FSCN may serve as a supplementary biomarker in cervical carcinogenesis, particularly in advanced lesions. Ma and Li demonstrated that FSCN1 is significantly upregulated in CC tissues, while its regulatory microRNA, miR-145, is markedly downregulated [115]. An inverse correlation between FSCN1 and miR-145 levels was observed, and luciferase reporter assays confirmed FSCN1 as a direct target of miR-145. Both miR-145 overexpression and FSCN1 silencing suppressed CC cell proliferation and colony formation. These findings suggest that the miR-145/FSCN1 axis plays a critical role in CC progression and highlights FSCN1 as a potential oncogenic effector regulated by tumor-suppressive miR-145 [115]. Similarly, He et al. also suggested that miR-145-5p suppresses CC progression by directly targeting FSCN1 [116]. Chetry et al. demonstrated that LGALS1 overexpression significantly promoted and enhanced the aggressive features of CC, which correlated with high FSCN expression [117].

In PIK3CA-mutant CC, Li et al. found that high FSCN1 expression predicts poor survival and radioresistance; silencing FSCN1 or its effector YWHAZ sensitized tumor cells to irradiation, revealing a synthetic lethal interaction [118]. Wang et al. demonstrated that decreasing FSCN1 expression resulted in decreased levels of β-catenin and C-myc RNA. This suggests that FSCN1 may enhance CC progression via this pathway [119]. Han et al. reported that lactic acid drives β-catenin nuclear translocation, redistributing FSCN to the cytoplasm and promoting migratory protrusions [120].

Guo et al. aimed to define the function of FSCN1 more broadly. They identified FSCN1 as a positive regulator of the angiogenic factor ANGPTL4, linking it to tumor angiogenesis [121]. Collectively, the above studies identify FSCN1 as a multifunctional oncogenic effector in CC, integrating HPV-, miRNA-, and signaling-driven pathways to promote migration, invasion, angiogenesis, and treatment resistance.

#### 3.3.7. Gelsolin

Gelsolin (GSN), which plays a role in many diseases, for example, Alzheimer’s disease, heart failure, or many types of cancer, is one of the most prevalent ABP [122]. This protein is responsible for cell mobility, programmed cell death, division, and phagocytosis. Its exact role in cancer remains unclear, although it has been noted that in many types of cancer, namely hepatocellular carcinoma, colon cancer, stomach cancer, and CC the levels of GSN are lowered [83]. Lee et al. reported that in CC tissues, USP7 and UHRF1 were upregulated, while GSN was downregulated. Silencing UHRF1 increased GSN expression, induced cell-cycle arrest, and promoted apoptosis, suggesting that GSN suppression via the USP7–UHRF1 axis supports tumor progression [122].

Among the molecular pathways subverted by HPV16 oncoproteins, the interaction between E7 and the actin binding protein GSN has emerged as a critical axis in the modulation of cytoskeletal dynamics and cell motility. In a comprehensive study by Matarrese et al., demonstrated that the E7 oncoprotein physically interacts with GSN, thereby promoting profound reorganization of the actin cytoskeleton, enhanced cellular motility, and features consistent with EMT [123]. Wild-type E7—but not GSN-binding-deficient mutants—promoted F-actin polymerization, vimentin upregulation, and E-cadherin loss, leading to increased invasion in SiHa and CaSki cells. E7 knockdown reversed these effects. A subsequent study showed that E7–GSN interaction also modulates Rho GTPases and suppresses the HIPPO pathway via cytoplasmic retention of inactive YAP, linking viral oncogene activity to cytoskeletal and transcriptional reprogramming [124]. In contrast, Wüstenhagen et al. found that although GSN interacts with the HPV16 L2 capsid protein, its depletion did not affect pseudovirus entry, indicating a minor role in early infection stages [125]. Collectively, these findings identify GSN as a multifunctional regulator in CC—suppressed in tumor tissues but exploited by HPV16 E7 to promote actin remodeling, EMT, and oncogenic signaling.

#### 3.3.8. Transgelin

Transgelin 2 (TAGLN2) belongs to actin- and calmodulin-binding protein family, which influences the structure and movement of the cell. This protein has been suspected to play a suppressing role in the process of metastasis. While members of the transgelin family can suppress metastasis through ERK1/2 modulation and MMP-9 downregulation, TAGLN2 has shown both tumor-suppressive and oncogenic roles depending on the cancer type [126].

Zhou et al. reported significantly reduced TAGLN2 mRNA and protein levels in CC. It correlates with advanced FIGO stage, poor differentiation, and shorter survival [126]. TAGLN2 overexpression in HeLa cells decreased viability, migration, and invasion, accompanied by increased E-cadherin and reduced CXCR4, MMP-2, and MMP-9 levels, suggesting suppression of NF-κB-related metastatic signaling [127].

Conversely, Yakabe et al. found that TAGLN2 knockdown inhibited migration and MMP secretion in SKG IIIa cells, implying a pro-tumorigenic function. Clinically, their data showed variable TAGLN2 expression across CC stages, with low expression predominating in advanced cases [128]. Discrepancies between studies likely stem from differences in disease stage and cell line models.

#### 3.3.9. Tropomodulin

Tropomodulins (TMOD) belong to the protein family responsible for capping the pointing end of actin filaments and modulating their depolymerization. As TMODs modulate actin dynamics, they were found to be linked to cell migration suppression and cancer progression [129,130]. Lu et al. demonstrated that TMOD1 knockdown in CC cell lines increased migration and invasion, effects reversed by TMOD1 re-expression. However, EMT markers (Twist, Snail) remained unchanged, suggesting an EMT-independent mechanism [129]. Analysis of clinical data further revealed that high TMOD1 expression correlated with early pathological stages, while low TMOD1 levels associated with advanced disease, supporting its role as a potential tumor suppressor in CC [129].
cells-14-01640-t005_Table 5Table 5Summary of studies analyzing actin-binding proteins (ABPs) in cervical cancer.Protein/TargetKey Role/Function in CCMain Mechanisms/PathwaysClinical/Experimental InsightsReferencesActinin 4(ACTN4)Promotes proliferation, migration, invasion, EMTWnt/β-catenin, PI3K/AKT/mTOR, Snail/MMP-9Overexpressed in CC tissues and serum; correlated with stage and metastasis; knockdown reduces stemness and invasion; potential diagnostic marker[74,75,76,77,78,79,80,81]Actinin 1(ACTN1)Regulates cytoskeleton and motilitymiR-129-5p targetingmiR-129-5p suppresses ACTN1, inhibiting migration, invasion, and angiogenesis[77]Cofilin 1(CFL1)Controls actin turnover, migration, invasionLIMK1/2–p-cofilin, ROS/Src, WWC2/YAP axisUpregulated in CC; phosphorylation regulates motility; LIMK and YAP signaling modulate activity; inhibitors reduce invasion[67,82,84,85,86,87,88]CortactinPromotes invasion, lesion progressionVEGF-C/c-Src ↓miR-326 → ↑Cortactin; cytoskeletal remodelingExpression increases with lesion grade; potential biomarker for progression[89,90]DIAPH3Enhances proliferation, tumor growthmTOR activation, modulation of immune cell infiltrationOverexpressed in CC; knockdown reduces proliferation and tumor size *in vitro* and *in vivo*[91,92,93]Ezrin (EZR)Supports migration, EMT, immune evasionPI3K/Akt signaling; scaffolding of CD47 and PD-L1Overexpression correlates with poor prognosis; knockdown reduces invasion and colony formation[85,94,95,96,97,98,99,100,101,102,103,104,105,106,107,110]Radixin (RDX)Promotes proliferation, invasionc-Jun/HIF1A-AS2/miR-34b-5p axisOverexpression associated with CC progression[108]Moesin (MSN)Regulates immune evasionMembrane localization of PD-L1Knockdown reduces PD-L1 surface expression[109]Fascin (FSCN1)Enhances motility, invasionFilopodia formation, miR-145/FSCN1 axis, Wnt/β-catenin, angiogenesis (ANGPTL4)Overexpressed in CC and HPV16+ lesions; correlates with poor prognosis and radiotherapy resistance[111,112,113,114,115,116,117,118,119,120]Gelsolin (GSN)Mediates cytoskeletal remodeling, EMTHPV16 E7 interaction → ↑F-actin, HIPPO pathway suppression; USP7/UHRF1 ↓GSNDownregulated in CC; knockdown reduces invasion; therapeutic target potential[82,121,122,123,124]Transgelin 2 (TAGLN2)Tumor suppressor↑E-cadherin, ↓MMP-2/MMP-9, NF-κB modulationDownregulated in advanced CC; overexpression inhibits migration and invasion[125,126,127]Tropomodulin 1 (TMOD1)Tumor suppressor, regulates actin dynamicsCapping actin filaments; controls motilityKnockdown ↑migration/invasion; high expression associated with early-stage CC[128,129]↓—lower level; ↑—higher level.

#### 3.3.10. Expression Patterns of ABP in CC

ABPs are crucial for cancer cell migration because, through reorganization of the actin cytoskeleton, they modulate the formation of migratory protrusions, contributing to tumor invasion. Overexpression of ABPs such as ACTN4, CFN, cortactin, DIAPH3, EZR, and FSCN is also observed in CC, contributing to increased metastasis and poorer patient prognosis. This hypothesis is supported by the reduced migration and invasion of cancer cells through reduced ABP expression. Specific signaling pathways and molecular interactions, for example, with HPV oncoproteins, miRNAs, or other proteins, have been identified that regulate ABP activity. This influences proliferation, EMT, and apoptosis evasion in CC cells. Summarizing the above section, ABPs represent an excellent therapeutic target for patients.

To complement the literature, a descriptive analysis of publicly available transcriptomic datasets was performed to map the expression of selected ABP-encoding genes (FSCN1, GSN, PFN1, PFN2, CTTN, DIAPH3) in normal and tumor cervical tissues. As illustrated in Figure 6 and Figure 7, FSCN1, PFN1, PFN2, and DIAPH3 were consistently upregulated in CC samples. GSN and CTTN showed inconsistent expression levels, which could be due to the quantity and quality of databases collected. These results collectively indicate that several ABPs are dysregulated in CC and may contribute to altered cytoskeletal dynamics during disease progression. (Figure 6 and Figure 7).

## 4. Summary and Perspectives

The reports cited in our article mapping the current state of knowledge regarding the role of selected integrin subunits, FA, FAK, and ABPs, in CC progression. These studies clearly indicate that cancer development proceeds in stages—from alterations in adhesion processes to full invasion—and is driven by disruptions in the integrin-FA/FAK-ABP system. The complexity of this signaling axis means that dysfunction of any of its components can contribute to increased aggressiveness and metastasis. Importantly, this knowledge has significant translational potential, creating new opportunities for the development of drugs, vaccines, and diagnostic biomarkers. Understanding the mechanisms regulating adhesion and invasion allows for the identification of specific therapeutic targets. For example, MMP inhibitors or EMT blockers can limit ECM degradation, thereby reducing the ability of cells to invade tissues [131,132]. FAK inhibitors, some of which are already being clinically studied, may in turn suppress integrin-dependent survival signaling and increase the sensitivity of tumor cells to radiotherapy and chemotherapy [133]. At the same time, diagnostics based on adhesion molecules (e.g., expression of integrin αvβ6 or vinculin) may be promising prognostic and predictive tools. In the long term, immunotherapies targeting adhesion-associated epitopes may complement the effects of current HPV vaccines by inhibiting tumor progression after infection.

However, significant research gaps remain. Only a few clinical studies have validated the prognostic utility of adhesion-related molecules, and data on less-studied components, such as ZYX or specific ABPs, remain limited. Moreover, as a scoping review, this study did not include formal quality assessment or quantitative synthesis, which limits the ability to draw definitive conclusions regarding effect sizes or clinical efficacy. Publication bias and the heterogeneity of experimental models also represent inherent limitations. Future research should focus on validating these molecular targets in the clinical setting and utilizing bioinformatics and proteomic analyses to identify key regulatory points in the adhesion-invasion axis. Combining molecular biology, pharmacology, and computational modeling approaches could accelerate the translation of these findings into clinical practice.

In summary, the integrin–FA/FAK–ABP signaling network constitutes a central mechanism underlying CC progression and a promising framework for identifying molecular targets and predictive biomarkers. From a clinical perspective, translating this knowledge into practical tools—such as specific pathway inhibitors, predictive tests, and new adjuvant therapies—could significantly improve the prognosis and treatment efficacy of patients with CC. This scoping review highlights the importance of ongoing interdisciplinary research to translate molecular insights into clinical applications that may enhance diagnosis, prognosis, and treatment of CC.

## Figures and Tables

**Figure 1 cells-14-01640-f001:**
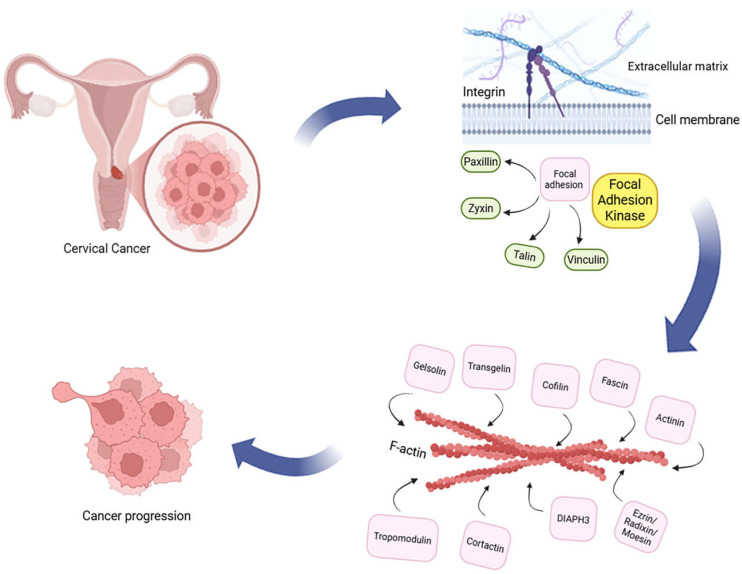
Schematic representation of the integrin–focal adhesion–actin-binding protein (Integrin–FA–FAK–ABP) axis in cervical cancer progression. Integrins mediate cell–extracellular matrix (ECM) interactions and initiate intracellular signaling through focal adhesion complexes composed of vinculin, talin, paxillin, and zyxin, as well as focal adhesion kinase (FAK). Activation of FAK triggers downstream pathways that regulate cytoskeletal remodeling via actin-binding proteins (ABPs) such as actinin, cofilin, fascin, gelsolin, cortactin, DIAPH3, and the ezrin/radixin/moesin complex. The coordinated activity of these molecules enhances cancer cell motility, invasion, and metastasis, contributing to cervical cancer progression. (Designed with BioRender: https://www.biorender.com/ accessed on 13 October 2025).

**Figure 2 cells-14-01640-f002:**
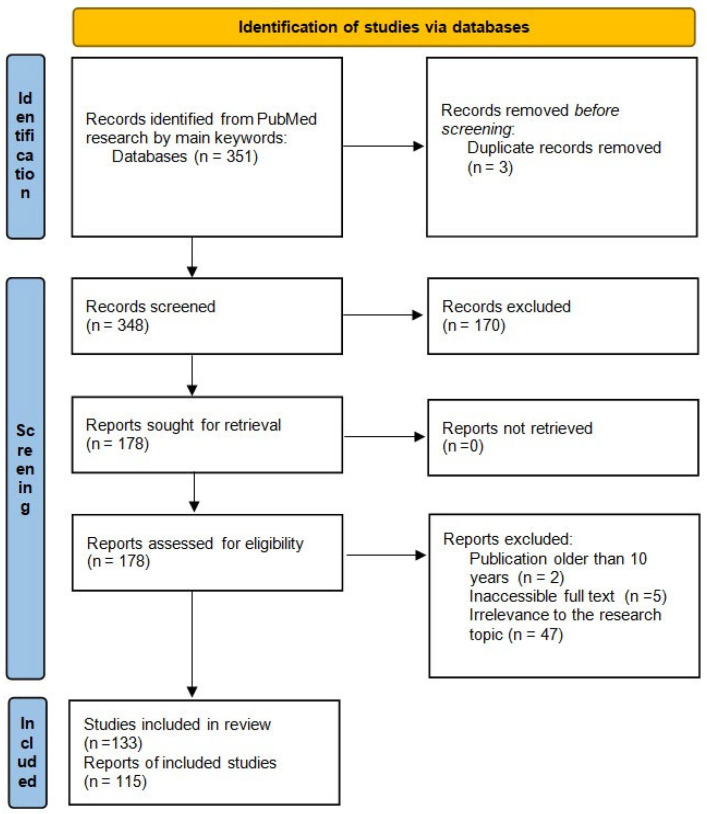
PRISMA flow diagram illustrating the article selection process for the scoping review. Records were excluded based on publication date (>10 years), duplication, inaccessibility of full text, and lack of relevance to the roles of integrins, FAs, FAK, and ABPs in CC progression [24]. n—number of publication.

**Figure 3 cells-14-01640-f003:**
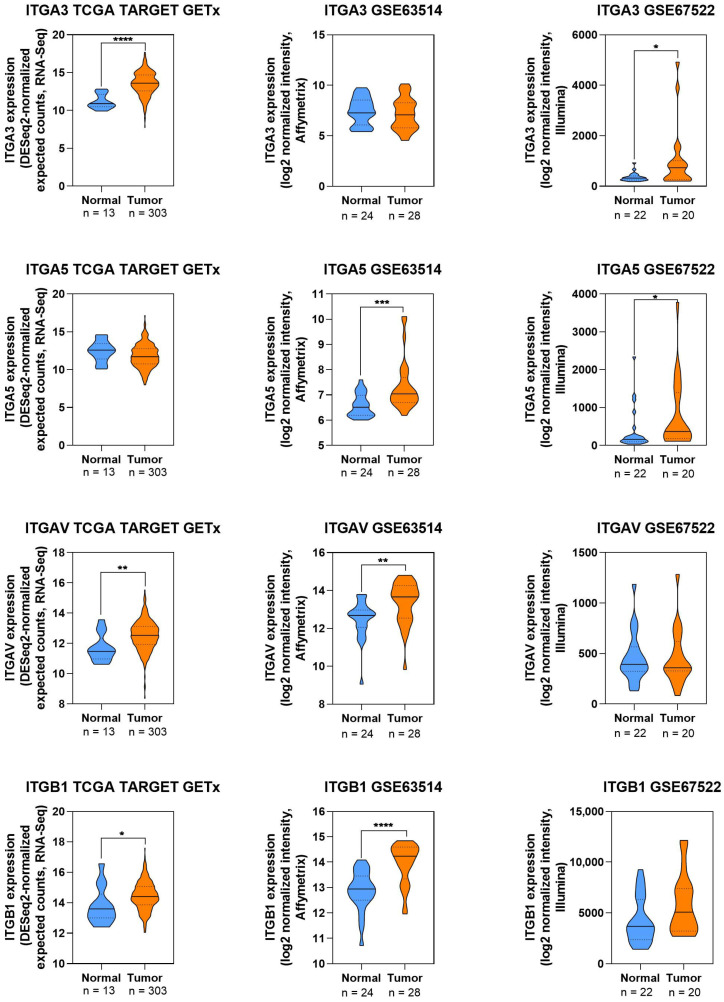
Expression profiles of integrin in normal and cervical cancer tissues. ITGA3—integrin α3; ITGA5—integrin α5; ITGAV—integrin V; ITGB1—integrin β1. Data were downloaded from the public databases TCGA, GSE63514, and GSE67522. Analyses included data from RNA-seq, Affymetrix, and Illumina. The integrated mapping supports literature evidence linking these molecules to migration, invasion, and angiogenesis. Statistical analysis was performed in GraphPad Prism 6 Software. Significance was presented in the figures as * *p* < 0.05, ** *p* < 0.01, *** *p* < 0.001, or **** *p* < 0.0001.

**Figure 4 cells-14-01640-f004:**
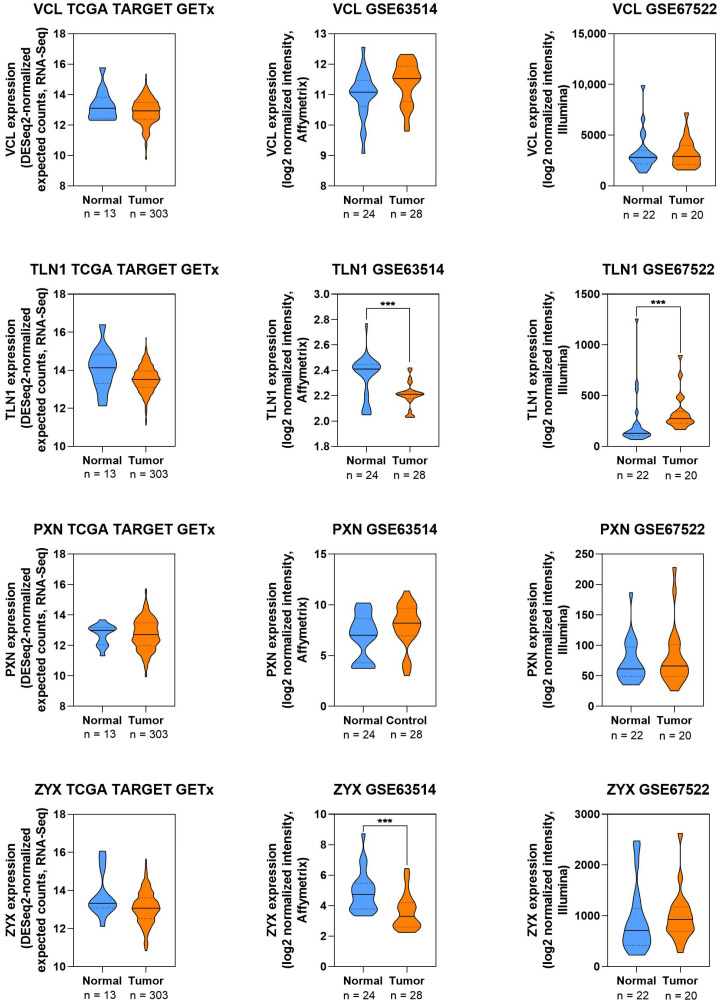
Expression profiles of FAs in normal and cervical cancer tissues. VCL—vinculin; TLN1—talin1, PXN—paxillin; ZYX—zyxin. Data were downloaded from the public databases TCGA, GSE63514, and GSE67522. Analyses included data from RNA-seq, Affymetrix, and Illumina. The integrated mapping supports literature evidence linking these molecules to migration, invasion, and angiogenesis. Statistical analysis was performed in GraphPad Prism 6 Software. Significance was presented in the figures as *** *p* < 0.001.

**Figure 5 cells-14-01640-f005:**
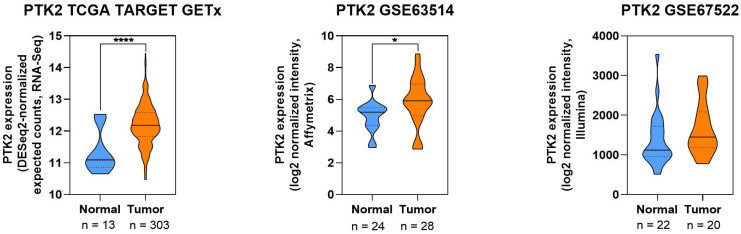
Expression profiles of FAK in normal and cervical cancer tissues. PTK2—protein tyrosine kinase 2. Data were downloaded from the public databases TCGA, GSE63514, and GSE67522. Analyses included data from RNA-seq, Affymetrix, and Illumina. FAK expression is consistently elevated in CC samples, suggesting its key role in regulating adhesion-mediated signaling and cancer progression. Statistical analysis was performed in GraphPad Prism 6 Software. Significance was presented in the figures as * *p* < 0.05, **** *p* < 0.0001.

**Figure 6 cells-14-01640-f006:**
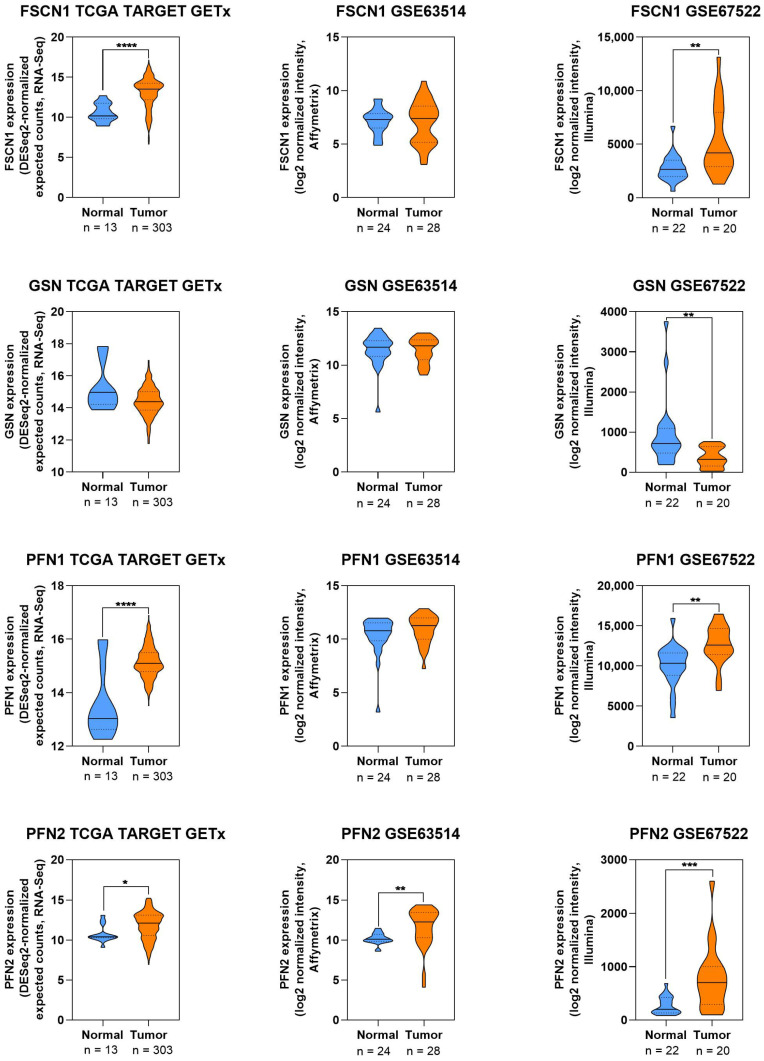
Expression profiles of selected ABPs in normal and cervical cancer tissues. FSCN1—fascin 1; GSN—gelsolin; PFN1—profilin 1; PFN2—profilin 2. Data were downloaded from the public databases TCGA, GSE63514, and GSE67522. Analyses included data from RNA-seq, Affymetrix, and Illumina. ABP expression highlights their association with the increased migratory potential of cancer cells. Statistical analysis was performed in GraphPad Prism 6 Software. Significance was presented in the figures as * *p* < 0.05, ** *p* < 0.01, *** *p* < 0.001, or **** *p* < 0.0001.

**Figure 7 cells-14-01640-f007:**
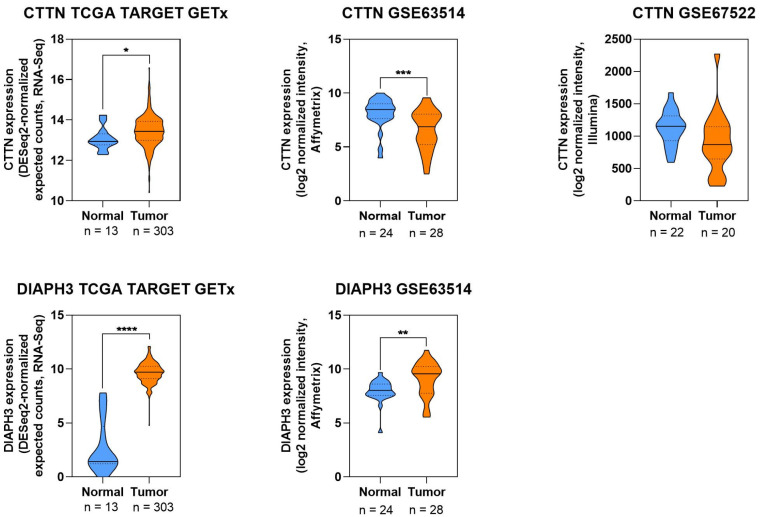
Expression profiles of selected ABPs in normal and cervical cancer tissues. CTTN—cortactin; DIAPH3—Diaphanous-related formin 3. Data were downloaded from the public databases TCGA, GSE63514, and GSE67522. Analyses included data from RNA-seq, Affymetrix, and Illumina. Statistical analysis was performed in GraphPad Prism 6 Software. Significance was presented in the figures as * *p* < 0.05, ** *p* < 0.01, *** *p* < 0.001, or **** *p* < 0.0001.

**Table 1 cells-14-01640-t001:** Proteins in CC progression and metastasis.

Protein Group	Examples	Main Function	Role in CC Progression/ Metastasis
Integrins	ITGA3, ITGA5, ITGB1, ITGAV	Receptors linking the cell to the ECM, initiating focal adhesion formation	Adhesion, migration, and pro-oncogenic signaling
Focal Adhesion proteins (FAs)	Paxillin, Vinculin, Zyxin, Talin	Structural and adaptor proteins—connect integrins with the cytoskeleton, stabilize focal adhesions	Cytoskeleton reorganization and cancer cell migration
Focal Adhesion Kinase (FAK)	FAK	Tyrosine kinase—central signaling hub in focal adhesions	Migration, proliferation, survival, and EMT
Actin binding proteins (ABPs)	Filamin, Gelsolin, Profilin, Actinins, Cortactin, Cofilin, Diaphanous-related formin 3, Fascin, Transgelin 2, Tropomodulin, Ezrin/Radixin/Moesin	Link FA and integrins with actin filaments, modulate cytoskeleton dynamics	Cell movement, invasion, and cytoskeleton remodeling

## Data Availability

The original contributions presented in the study are included in the article, further inquiries can be directed to the corresponding author.

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
