# Peer review of "From Adhesion to Invasion: Integrins, Focal Adhesion Signaling, and Actin Binding Proteins in Cervical Cancer Progression—A Scoping Review"

_cells, 2025, doi:10.3390/cells14201640_

Round 1
Reviewer 1 Report
Comments and Suggestions for Authors
The present manuscript (ID:cells-3884784) titled "From adhesion to invasion: integrins, focal adhesion signaling, and actin binding proteins in cervical cancer progression" highlights that cervical cancer progression is a stepwise process from viral adhesion to cellular invasion, and better knowledge of these mechanisms can guide diagnosis, prognosis, and targeted treatment. The main weaknesses of the manuscript lie in its descriptive rather than critical approach, with sections often summarizing known information without sufficient comparison, analysis, or highlighting of controversies.
Abstract. The authors should add one sentence that explains the novel contribution (e.g., “This review uniquely links adhesion molecules with invasion mechanisms in cervical cancer progression”). End with a strong translational message, for example “Understanding these pathways may support biomarker discovery and targeted therapies".
Introduction. The authors should mention why adhesion-to-invasion is a critical concept in cervical cancer. Most of the references are outdated. The authors should include "2023–2025" references to strengthen the introduction.
3. Main Body (Adhesion, Invasion Mechanisms, EMT, etc.). Flow between subsections is abrupt, making it hard to see how adhesion leads stepwise to invasion.
The authors should include summary tables comparing adhesion molecules, EMT markers, and signaling pathways.
Discussion. The authors should discuss how adhesion/invasion knowledge can be used in drug or vaccine development. Also expand future directions with concrete examples (e.g., MMP inhibitors, EMT blockers, adhesion molecule–based diagnostics).
Conclusion. It is very short. and does not highlight the most important message of the paper. Reframe to emphasize the central finding: “Cervical cancer progression follows a stepwise process from adhesion to invasion, with multiple molecular targets for therapy and prognosis.” The authors should add a final forward-looking sentence on translational applications.
Author Response
Dear Reviewer,
We sincerely thank the reviewer for their valuable comments and constructive feedback, which helped us improve the quality and clarity of our manuscript.
We have attached our responses to all suggestions in the file below.
We believe that the implemented revisions have substantially strengthened the manuscript and enhanced its overall quality.
Thank you once again for your valuable feedback and the opportunity to revise our work.
Kind regards,
Marta Hałas-Wiśniewska

Reviewer 2 Report
Comments and Suggestions for Authors
Overview of the manuscript
The work is a review that focuses on the involvement of integrins and several adhesion and cytoskeletal factors in cervical cancer progression and metabolic processes. In addition to a bibliographic analysis, the authors perform a statistical comparison of data derived from available databases. They conclude that understanding the interactions between adhesion and cytoskeletal factors may represent an important step toward developing more effective anticancer therapies and improving prognosis.
GENERAL COMMENT
The work is very interesting and presents, in the statistical comparison, a specific point of interest. The review is rich in data, supported by a wide analysis of the literature. However, I suggest reducing the section related to integrins, as this is a well-known topic that does not add significant innovative information. Furthermore, summary tables should be included for several sections to make it easier to consult the relevant facts. Some points should be clarified more clearly. The bibliographic references are appropriate and sufficiently up to date.
Specific comments
Page 4–7: The paragraphs describing individual integrins should be shortened, keeping only the most relevant and innovative information.
Page 9, lines 332–339: This section should be moved to the Introduction. Leave here only a few introductory sentences.
Page 15, lines 755–757: The sentence is too long and unclear. Please rephrase it.
References
References 45: the year of publication is missing
Author Response
Dear Reviewer,
We sincerely thank the Reviewer for his valuable comments and constructive feedback, which helped us improve the quality and clarity of our manuscript.
We have attached our responses to all suggestions in the file below.
We believe that the revisions significantly strengthened the manuscript and improved its overall quality.
Thank you again for your valuable comments and the opportunity to improve our work.
Sincerely,
Marta Hałas-Wiśniewska

Reviewer 3 Report
Comments and Suggestions for Authors
The authors prepared a comprehensive review on role of integrins, focal adhesion signaling, and actin binding proteins in cervical cancer progression.
- The manuscript contains five figures. These figures present a statistical analysis of the expression of various proteins in cervical tumor tissue. Authors are encouraged to include additional illustrations schematically demonstrating the involvement of these protein factors in cervical cancer progression.
- Introductory sections 3, 4 and 5 need to be expanded to explain why, for example, in cervical cancer progression, researchers focus on these specific factors described in the review. Specifically, approximately 19 alpha- and 8 beta-integrins are currently known. The authors describe only three types of integrins.
- The review is replete with details, some of which could have been shortened without compromising the presentation. For example, the research methods are described in excessive detail, like “After 1 hour, unattached cells were counted” (lines 396-397). “Changes in VCL expression were also assessed.” (line 402), or some phrases are overly general (line 441).
- Check all abbreviations. Some abbreviations (ERK, JNK, IHC, MMP, LASSO, COX etc) are used without explanation. Abbreviations should be used in same format (see COX vs Cox, line 284). Abbreviations cannot be used as keywords (line 29).
- Some paragraphs are very long and should be split into subsections (see lines 221-262).
- Check the font size in the legend to Figure 1 (lines 325-330).
Author Response

(The authors gave the same response as above.)

Reviewer 4 Report
Comments and Suggestions for Authors
In this review article, the authors focused on highlighting the roles of the main integrins, focal adhesion proteins (Fas), and actin-binding proteins (ABPs) in the context of cervical cancer.
I appreciated the way they described the database search parameters; it's rare to find this type of information in such detail. This is important since much of the information in this review article (especially the figures created by the authors) was obtained from these databases. Specifically regarding Fas and ABPs, the authors were able to gather the most relevant information about the function of these proteins and their possible links to tumor progression.
Conceptually, the definition of integrins should include not only outside-in signaling, but also inside-out signaling, as previously described in many studies. Throughout the text, the authors give the impression that integrins are only involved in signaling from the external environment (see lines 14 and 40).
Another factor that may cause some confusion is that the authors treat the integrin subunits as if they were complete receptors, leading the reader to interpret them this way. Examples can be found primarily in section 3.4. On the other hand, there is no commentary on the possible partners of each alpha or beta subunit in the effects described.
I found no information in the text about the possible functional differences between the integrins. Focal adhesion proteins (FAs) and ABPs are relatively common to focal adhesions mediated by any integrin, and one would not expect many differences between them. To date, as far as is known, they have not been proposed as molecular targets for treatment precisely because of their promiscuity.
Author Response

(The authors gave the same response as above.)

Round 2
Reviewer 1 Report
Comments and Suggestions for Authors
The authors have responded to my comments in the revised manuscript.
Comments on the Quality of English LanguageThe authors have responded to my comments in the revised manuscript. Please feel free to take your final decision.